# Testosterone Nanoemulsion Prevents Prostate Cancer: PC-3 and LNCaP Cell Viability In Vitro

**DOI:** 10.3390/ijms25147729

**Published:** 2024-07-15

**Authors:** Marco Antonio Botelho, Dinalva Brito Queiroz

**Affiliations:** 1Ceara Institute of Science and Technology (IFCE), Sobral 62042-030, Brazil; 2Laboratoire Matière et Systèmes Complexes, UMR 7057, CNRS and Université Paris Diderot, 10 rue Alice Domon et Léonie Duquet, 75205 Paris cedex 13, France

**Keywords:** testosterone, prostate cancer, nanoparticles, nanoemulsion, nanotechnology

## Abstract

For many years, it has been speculated that elevated testosterone levels may be critically involved in the genesis and proliferation of prostate cancer. Methods: The effect of testosterone on the metabolic activity of hormone-independent [PC-3] and hormone-dependent [LNCAP] cancer cells was investigated in vitro. Additionally, the impact of testosterone nanoemulsion [nanocare^®^] on cell viability was accessed by flow cytometry. Results: Despite the dependency of the normal prostate and of most prostatic cancers upon androgens, the obtained results indicate that, contrary to prevailing opinion, the supplementation of testosterone with higher doses in nanoemulsion was able to lower the metabolic activity and viability of prostate cancer cells. Conclusions: We conclude that the growth of hormone-independent and hormone-dependent prostate cancer cells was reduced by the exposure of a nanoemulsion of bioidentical testostosterone in vitro. To the best of our knowledge, this is the first time that the potential effect of a testosterone nanoemulsion on the metabolic activity of prostate cancer cells has been shown. Such tests suggest that the growth of hormone-independent and hormone-dependent prostate cancer cells was reduced by the administration of bioidentical testostosterone, and this might be an interesting strategy for prostate cancer treatment in diagnosed patients.

## 1. Introduction

Prostate cancer [PCa] is one of the most frequent cancer diagnoses throughout the world. According to the American Cancer Society, in 2017, there was over 161,360 new cases of PCa diagnosed in the US, resulting in about 26,730 deaths [1].

Testosterone plays important roles in the human body, as it stimulates matrix synthesis by osteoblasts, stimulates osteoblast proliferation and differentiation, and prevents bone loss and calcium accumulation in soft tissues [2,3,4]. Prostate microcalcifications and subsequent lesions generally occur in men with low levels of testosterone. Recent studies have shown that low serum testosterone levels are predictive of prostate cancer [5]. The etiology of PCa represents a puzzling issue. Besides the strong correlation with low levels of testosterone, there is an increased relative risk for prostate calcification in patients with low serum levels of testosterone [3]. There is a significantly higher incidence for men with low levels of androgens [6].

Environmental risk factors comprise the consumption of sugar, wine, beer, and carbohydrates [7], as well as vitamin D deficiency [8]. Smoking may also induce an increased risk for PCa [9].

Inflammatory factors such as stress and obesity have been implicated in the development of PCa [10,11].

It is the prevailing opinion that the stimulation of prostatic tissue growth is induced by low androgen levels in aged human males, leading to an increased risk of prostate cancer [12].

It is certainly true that the prostate will not develop with high levels of testosterone, and the gland will experience atrophy if androgen supplementation is not provided [13]. In some animal models, prostate cancer can be produced and/or accelerated by the administration of different types of drugs that block the testosterone protection of this gland [14].

Nanotechnology has become a part of different fields, bringing new strategies and perspectives, especially in the medical sciences [15,16,17,18]. New, strong evidence suggests that nanodrugs have interesting and unique properties [19].

The aim of the present study was to determine the in vitro effect of a novel nanoemulsion (Nanocare^®^) of testosterone on prostate cancer cells. LNCAP [lymph node carcinoma of the prostate] and PC-3 [prostate adenocarcinoma] cells were treated with different concentrations of testosterone, and cell metabolic activity was measured by the Alamar Blue Test, while cell viability was accessed by flow cytometry.

## 2. Results

In this study, the effect of testosterone on the metabolic activity of hormone-independent [PC-3] and hormone-dependent [LNCAP] cancer cells was investigated in vitro.

The LNCaP cell line is derived from human prostate adenocarcinoma cells from a lymph node metastasis. They are adherent epithelial cells that grow in aggregates, as well as in single cells, and were initially obtained from a 50-year-old Caucasian male in 1977, where cells were taken from a needle aspiration biopsy of a metastatic lesion in the left supraclavicular lymph node. The LNCaP cell line is sensitive to hormones such as estrogen and androgen, which can be used to modulate growth. In fact, highly sensitive androgen receptors are present in the cytosol of LNCaP both in culture and in tumors, making LNCaP a highly androgen-sensitive cell line. Cells grown in culture and in prostate tumors both secrete phosphatase.

The cell line used in this study was stable, and the qualities of malignancy were maintained, making LNCaP useful for oncology purposes, such as investigating prostatic adenocarcinoma activity.

The differentiation was maintained, evidenced by the expression of Prostate-Specific Antigen [PSA] and Human Prostatic Acid Phosphatase [hPAP], indicating the presence of active cytosolic androgen receptors in prostate cells [20].

The PC-3 cell line is an epithelial cell line [PC-3] derived from a human prostatic adenocarcinoma metastatic to bone. The cultured cells show anchorage-independent growth in both monolayers and in soft agar suspension and produce subcutaneous tumors in nude mice. Culture of the transplanted tumor yielded a human cell line with characteristics identical to those used initially to produce the tumor. PC-3 does not respond to androgens, glucocorticoids, or epidermal or fibroblast growth factors. Karyotypic analysis by quinacrine banding revealed the cells to be completely aneuploid, with a modal chromosome number in the hypotriploid range. At least 10 distinctive marker chromosomes were identified. These cells are useful in investigating the biochemical changes in advanced prostatic cancer cells and in assessing their response to chemotherapeutic agents [21].

In order to investigate the effect of androgen on prostate cancer cell viability, PC3 and LNCaP cells were incubated with 62.5 to 250 µM testosterone. Non-treated cells were used to assemble a control group.

According to the microscopic evaluation [Figure 1], changes were observed when PC3 cells were incubated with testosterone at 62.5 µM. A reduced PC3 cell density was observed when the testosterone concentration was increased to 250 µM. Testosterone’s effect induced a higher impact for the LNCaP cells [Figure 2].

A reduction in cell density was observed both at 62.5 and 250 µM. In order to quantify these effects, the metabolic activity of PC3 and LNCaP cells after incubation with testosterone from 62.5 to 4000 µM concentration was evaluated by the Alamar Blue Test. Cell health can be monitored by numerous methods.

Plasma membrane integrity, DNA synthesis, DNA content, enzyme activity, the presence of ATP, and cellular-reducing conditions are known indicators of cell viability and cell death. The Alamar Blue cell viability reagent functions as a cell health indicator by using the reducing power of living cells to quantitatively measure the proliferation of various human and animal cell lines, bacteria, plants, and fungi, allowing one to establish the relative cytotoxicity of agents. When cells are alive, a reducing environment is maintained within the cytosol of each cell.

Resazurin, the active ingredient of the Alamar Blue reagent, is a non-toxic, cell-permeable compound that is blue in color and virtually non-fluorescent. Upon entering cells, resazurin is reduced to resorufin, a compound that is red in color and highly fluorescent. Viable cells continuously convert resazurin to resorufin, increasing the overall fluorescence and color of the media surrounding cells.

According to data from our Alamar Blue Test, a dose-dependent effect was observed for the impact of testosterone on PC3 metabolic activity [Figure 3].

For instance, at a 4000 µM testosterone dose, the PC3 metabolic activity was 14%. The metabolic activity was 24%, 83%, and 101% for testosterone doses of 1000, 250, and 62.5 µM, respectively.

A similar effect was observed for the LNCaP cells [Figure 4].

For example, at a 4000 µM testosterone dose, the LNCaP metabolic activity was 11.5%. The metabolic activity was observed to be 11%, 18%, and 33% for testosterone doses of 1000, 250, and 62.5 µM, respectively.

PC-3 cell viability was investigated by flow cytometry using annexin FITC and propidium iodide staining. Flow cytometry measures and analyzes different characteristics of cells. The properties measured included cell size and granularity and fluorescence intensity. These data provide information about subpopulations within the sample [22]. Herein, flow cytometry analysis was used to investigate cell viability via propidium iodide and annexin staining. One of the hallmarks of apoptosis is phosphatidylserine translocation from the inner to the outer leaflet of the cell membrane. Annexin is able to bind to phosphatidylserine in the outer leaflet of the cell membrane. Therefore, positive staining for annexin is indicative of apoptosis. Propidium iodide is a fluorescent molecule that binds to cell nuclei. The membrane of viable cells is impermeable to propidium iodide. Therefore, positive propidium iodide staining indicates irreversible cell membrane damage and cell death. According to annexin/propidium iodide staining, the cells were classified into four categories: dead but non-apoptotic cells [annexin-V-negative and propidium iodide-positive], late apoptotic cells [positive for both annexin-V and propidium iodide], cells at the onset of apoptosis [stained positive for annexin-V alone], and viable cells [negative for both annexin-V and propidium iodide] [23].

Propidium iodide × Annexin V staining dual parameter scatter plots are presented in Figure 5, showing that the cells shifted from being viable to early apoptotic and late apoptotic cells as the testosterone concentration increased. In fact, there was a statistically significant decrease [*p* = 0.023] in the percentage of viable cells from 56.64 ± 11.23% for the control group to 25.15 ± 10.31% at 250 µM testosterone concentration [Figure 6].

A non-statistically significant increase in the percentage of early apoptotic cells [Figure 7] from 37.6 ± 8.2 to 60.1 ± 11.8 was observed when comparing the control group to the 250 µM testosterone group. However, a statistically significant [*p* = 0.011] increase in the percentage of late apoptotic cells was induced by testosterone at a 250 µM concentration. The percentage of late apoptotic cells shifted from 4.6 ± 2.7 for the control to 12.7 ± 1.6 for the 250 µM testosterone group [Figure 8]. The percentage of necrotic cells was not impacted by testosterone, remaining between 1 and 3% for both the control group and the cells treated with testosterone [Figure 9].

## 3. Discussion

Microscopic analysis and, to a greater extent, the Alamar Blue test indicated that testosterone induced a dose-dependent cytotoxic effect for the PC3 cells. The same applied to the LNCaP cells but to a greater extent than observed for the PC3 cells. The impact of testosterone on cell viability was further proved by annexin/propidium iodide analysis by flow cytometry for the PC3 cells. These results seem contradictory to literature data. For instance, Antognelli et al. showed a near 5-fold proliferation increment in the presence of testosterone at a 100 nM concentration for LNCaP cells, while a no change in cell proliferation at the same testosterone concentration was observed for the PC3 cells [24]. In their experimental set-up, Antognelli et al. evaluated proliferation by the [^3^H] thymidine incorporation assay, which is a different protocol compared to the methodology investigated herein. However, the most important point to mention is that our results concern a much higher testosterone level. While Antognelli et al. evaluated testosterone’s effect within the nM range, our experiments were carried out within a µM range, corresponding to the effects of testosterone at much higher doses.

## 4. Materials and Methods

### 4.1. Cell Culture

The PC-3 cells [ATCC CRL-1435] were cultured in DMEM medium supplemented with 10% [*v*/*v*] fetal bovine serum, 100 U/mL penicillin–streptomycin, and 2 mM L-glutamine at 37 °C in 5% CO_2_. The LNCaP cells [ATCC] were cultured in RPMI medium supplemented with 10% [*v*/*v*] fetal bovine serum, 100 U/mL penicillin–streptomycin, and 2 mM L-glutamine at 37 °C in 5% CO_2_. The cells were passed twice a week once they reached 90% confluence.

### 4.2. Cell Metabolic Activity Evaluation—Alamar Blue Test

PC3 and LNCaP cells were plated in 98-well plates. The cells were incubated with testosterone in complete medium overnight at 0, 62.5, 125, 250, 500, 1000, and 4000 µM concentrations. The cells were rinsed twice and incubated with complete medium for 24 h. The cells were observed using an optical microscope [Leica, Wetzlar, Germany], and their metabolic activity was assessed by the Alamar Blue Test [Invitrogen, Carlsbad, CA, USA], carried out according to the supplier’s instructions. The cells were incubated with 10% Alamar Blue for two hours. The fluorescence in the cell medium due to the reduction of resazurin [oxidized form] to resorufin by cell activity was quantified on a FLUOstar OPTIMA microplate reader [excitation 550 nm, emission 590 nm].

### 4.3. Cell Viability Evaluation—Flow Cytometry

PC3 cells were plated onto flasks of 25 cm^2^. The cells were incubated with testosterone in complete medium overnight at 0, 62.5, 125, and 250 µM concentrations. The cells were rinsed twice and incubated with complete medium for 24 h.

The PC-3 cells were suspended in binding buffer and incubated with Annexin-V FITC conjugate and propidium iodide solution at room temperature for 10 min and protected from light, according to the manufacturer’s protocol [Annexin-V-Fluos kit, Roche]. The cells [10,000 events] were then analyzed using the CyAn ADP LX flow Cytometer [Beckman Coulter, San Jose, CA, USA] on the Platform ImagoSeine, IJM, Université Paris Diderot, France.

Populations of viable cells [negative for both annexin-V and propidium iodide], dead but non-apoptotic cells [annexin-V negative and propidium iodide positive], and late apoptotic cells [positive for both annexin-V and propidium iodide], as well as cells at the onset of apoptosis [stained positive for annexin V alone], were quantified separately on a biparametric dot plot representing propidium iodide intensity versus FITC fluorescence intensity.

### 4.4. Particle Size Z-Average and Physical Stability

Particle size analysis was performed by dynamic light scattering, also known as photon correlation spectroscopy, using a particle size analyzer (Zetasizer Nanoseries-ZS90 (Malvern, UK)).

Prior to the measurements, all samples were diluted (1:360) using Milli-Q water to yield a suitable scattering intensity. Dynamic light scattering data were analyzed in disposable sizing cuvettes at a laser wavelength of 633 nm, 25 °C, with a fixed light incidence angle of 90°. The mean hydrodynamic diameter (Z-average) and the polydispersity index were determined as a measure of the width of the particle size distribution. The Z-average and polydispersity index of the analyzed samples were obtained by calculating the average of 13 runs.

Measurements were performed in triplicate. The mean particle diameter of the testosterone nanoparticles was 232 nm. The nanoemulsion of testosterone presented a high negative average zeta potential of −42.8 mV. The physical stability of the testosterone nanoparticles was also evaluated by examining changes in mean particle sizes during storage for 2 months at room temperature. The testosterone nanoparticles did not show statistically significant changes in their mean diameter (*p* > 0.05) when stored at room temperature for 2 months.

### 4.5. Nanoemulsion Preparation

Testosterone was purchased from Sigma Aldrich. The nanoemulsion formulation of testosterone with the penetration enhancer (Nanocare^®^) was prepared in the Laboratory of Chemistry Technology at the Ceará Institute of Technology (IFCE).

### 4.6. Statistical Analysis

Student’s *t* test was performed to determine significant differences between the test and control groups using the Prism 3.0 version of GraphPad software [USA]. A minimum of a 95% confidence level was considered significant. *, **, and *** indicate *p* < 0.05, *p* < 0.01, and *p* < 0.001.

## 5. Conclusions

The results showed that administering testosterone at high concentration levels [especially at 250 µM concentration or higher] induced a dose-dependent cytotoxic effect in vitro on LNCAP and PC3 prostate cancer cells, according to the Alamar Blue Test.

Additionally, annexin/propidium iodide analysis by flow cytometry for the PC3 cells indicated that at a 250 µM concentration, testosterone enhanced early apoptosis and, to a greater extent, late apoptosis.

This suggests that therapy with testosterone may be beneficial for both the prevention and coadjuvant treatment of prostate cancer by reducing the viability of cancer cells. Further studies should be performed in vivo to confirm the results obtained in vitro.

## Figures and Tables

**Figure 1 ijms-25-07729-f001:**
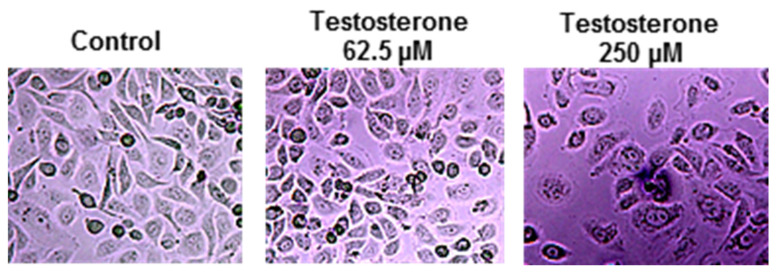
Microscopic observation of PC3 cells after the incubation with testosterone at 62.5 and 250 µM concentration levels compared to control.

**Figure 2 ijms-25-07729-f002:**
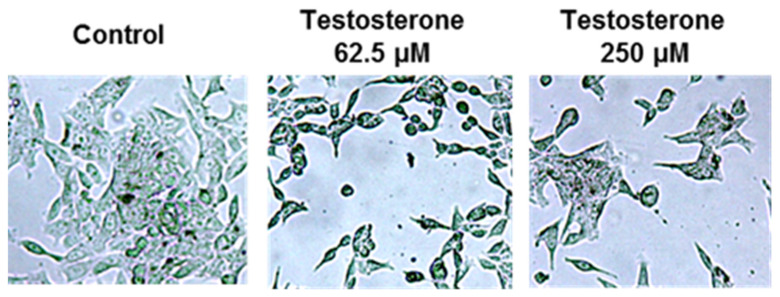
Microscopic observation of LNCaP cells after incubation with testosterone at 62.5 and 250 µM concentrations compared to control.

**Figure 3 ijms-25-07729-f003:**
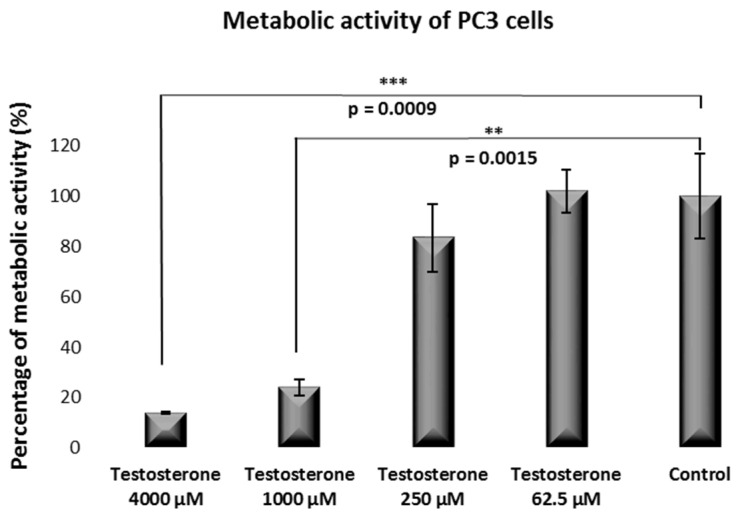
Metabolic activity of PC3 cells after incubation with testosterone from 62.5 to 250 µM concentrations compared to control.

**Figure 4 ijms-25-07729-f004:**
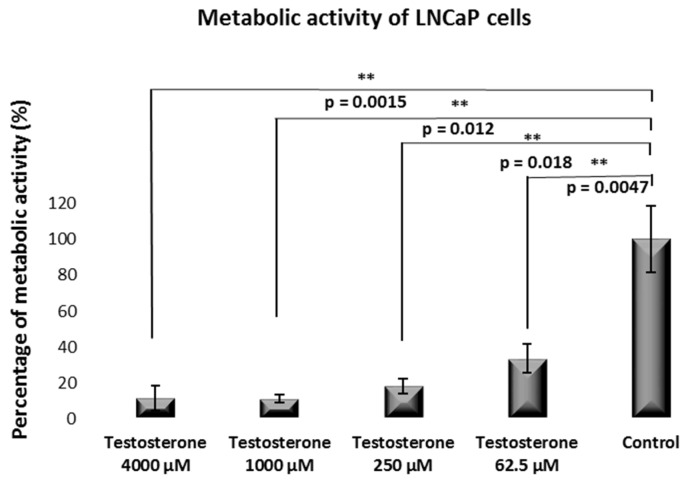
Metabolic activity of LNCaP cells after incubation with testosterone from 62.5 to 250 µM concentrations compared to control.

**Figure 5 ijms-25-07729-f005:**
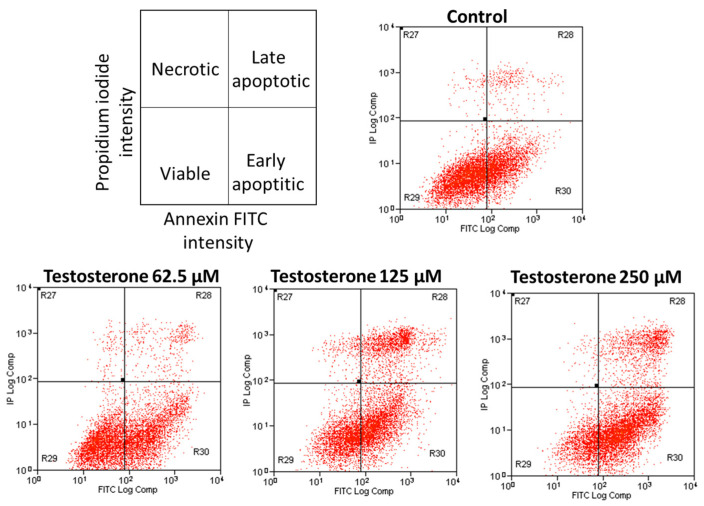
Flow cytometry dot plots for investigating viability of PC-3 cells incubated with testosterone from 62.5 to 250 µM compared to control. Propidium iodide × Annexin V staining dual parameter scatter plots are presented to help determine viable, early apoptotic, late apoptotic, and necrotic cell populations.

**Figure 6 ijms-25-07729-f006:**
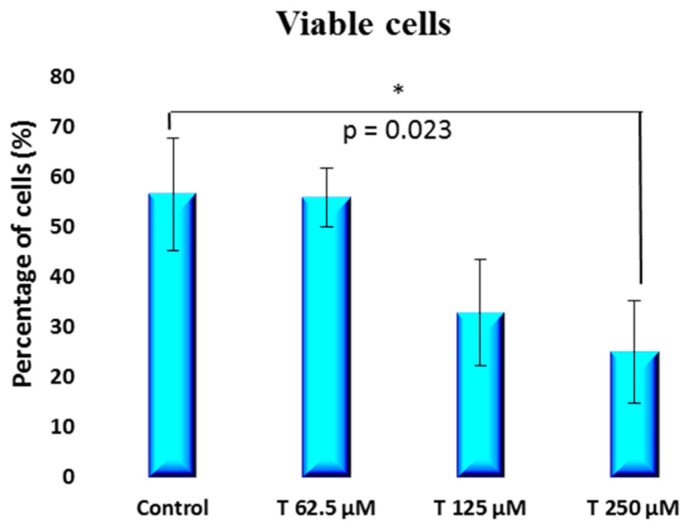
Percentage of viable cells obtained from flow cytometry dot plots for PC-3 cells incubated with 62.5 to 250 µM testosterone compared to control.

**Figure 7 ijms-25-07729-f007:**
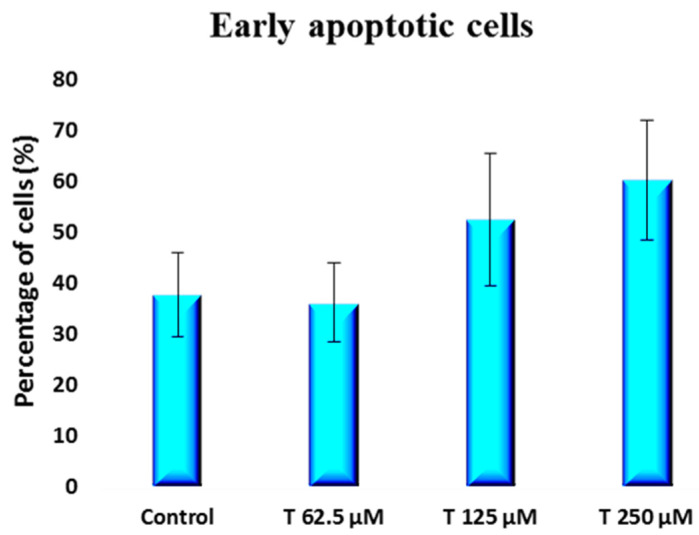
Percentage of early apoptotic cells obtained from flow cytometry dot plots for PC-3 cells incubated with 62.5 to 250 µM testosterone compared to control.

**Figure 8 ijms-25-07729-f008:**
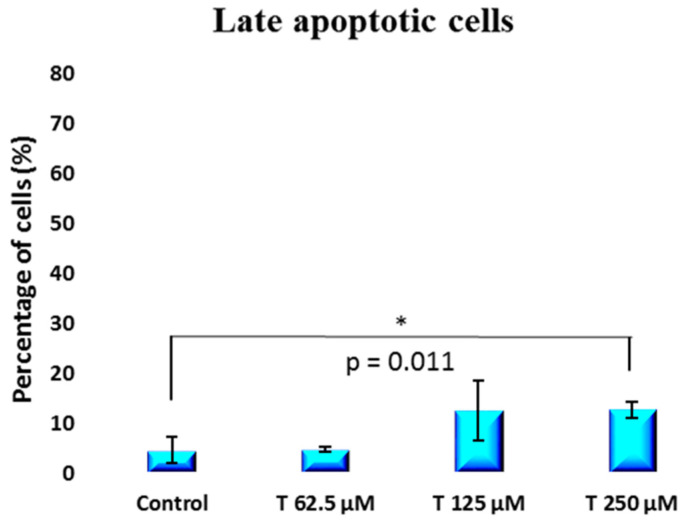
Percentage of late apoptotic cells obtained from flow cytometry dot plots for PC-3 cells incubated with 62.5 to 250 µM testosterone compared to control.

**Figure 9 ijms-25-07729-f009:**
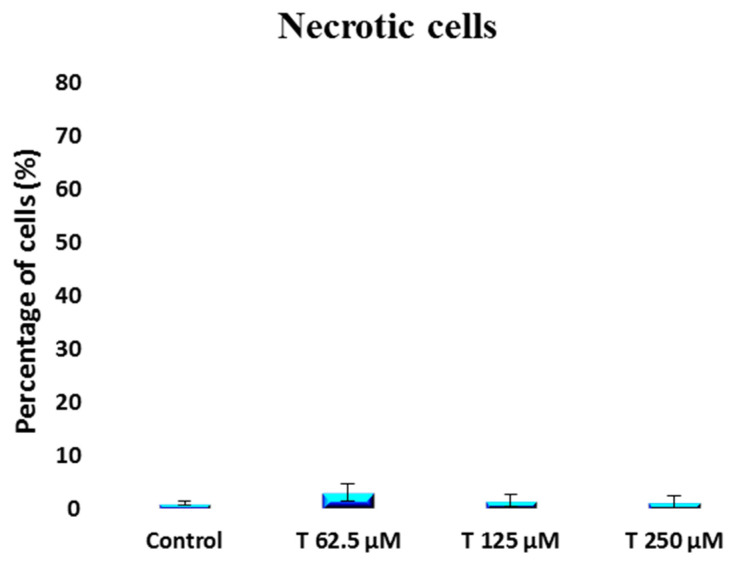
Percentage of necrotic cells obtained from flow cytometry dot plots for PC-3 cells incubated with 62.5 to 250 µM testosterone compared to control.

## Data Availability

The original contributions presented in the study are included in the article, further inquiries can be directed to the corresponding author.

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
