# Peer review of "Testosterone Nanoemulsion Prevents Prostate Cancer: PC-3 and LNCaP Cell Viability In Vitro"

_ijms, 2024, doi:10.3390/ijms25147729_

Round 1
Reviewer 1 Report
Comments and Suggestions for Authors
The manuscript reported that the supplementation of testosterone in nanoemulsion may lower the metabolic activity and viability of cancer cells. The result suggest that the growth of hormone-independent and hormone-dependent prostate cancer cells was reduced by the administration of bioidentical testostosterone.
1. The present conclusion were based on the Alamar blue test and annexin/propidium iodide analysis. It is very important to give a trustworthy conclusion for the cause of prostate cancer. I think more data should be provided to verify the authors conclusion.
2. The error bars were large in Figures 3, 5&6. Would the authors like to give an explanation?
3. There are some unrelated sentences which should be deleted, for example, “The current state of the research field should be carefully reviewed and key publications cited. Please highlight controversial and diverging hypotheses when necessary. Finally, briefly mention the main aim of the work and highlight the principal conclusions. As far as possible, please keep the introduction comprehensible to scientists outside your particular field of research. References should be numbered in order of appearance and indicated by a numeral or numerals in square brackets—e.g., [1] or [2,3], or [4–6]. See the end of the document for further details on references.” and “This section may be divided by subheadings. It should provide a concise and precise description of the experimental results, their interpretation, as well as the experimental conclusions that can be drawn.”.
Comments on the Quality of English LanguageEnglish language is good.
Author Response
For research article
Testosterone Nanoemulsion prevents Prostate Cancer: PC-3 and LNCaP Cell Viability in Vitro
|
Response to Reviewer X Comments
|
||
|
1. Summary |
|
|
|
Thank you very much for taking thetime to review this manuscript. Please find the detailed responses below and the correspondingrevisions/corrections highlighted/in track changes in the re-submitted files.
|
||
|
2. Questionsfor General Evaluation |
Reviewer’s Evaluation |
Response and Revisions |
|
Does the introduction provide sufficient background and include all relevant references? |
Yes/Can be improved/Must be improved/Not applicable |
|
|
Are all the cited references relevant to the research? |
Yes/Can be improved/Must be improved/Not applicable |
|
|
Is the research design appropriate? |
Yes/Can be improved/Must be improved/Not applicable |
|
|
Are the methods adequately described? |
Yes/Can be improved/Must be improved/Not applicable |
|
|
Are the results clearly presented? |
Yes/Can be improved/Must be improved/Not applicable |
|
|
Are the conclusions supported by the results? |
Yes/Can be improved/Must be improved/Not applicable |
|
|
3. Point-by-point response to Comments and Suggestions for Authors |
||
|
Comments 1: to Reviewer 1 The manuscript reported that the supplementation of testosterone in nanoemulsion may lower the metabolic activity and viability of cancer cells. The result suggest that the growth of hormone-independent and hormone-dependent prostate cancer cells was reduced by the administration of bioidentical testostosterone.
Response: The authors agreed and we gave a trustworthy conclusion for the cause of prostate cancer.
Response: The authors agreed and made the req2uested corrections on the manuscript
Resposnse to the reviewer 2
The manuscript entitled "Testosterone Nanoemulsion prevents Prostate Cancer: PC-3 and LNCaP Cell Viability in Vitro" is an interesting research work that paves the way for new possible therapeutic strategies in the treatment of prostate cancer. With this study the authors demonstrate that, contrary to what has been hypothesized so far, the supplementation of nanoemulsified testosterone leads to a reduction in the metabolic activity and viability of prostate cancer cells. However, I would suggest to the authors a small modification of the different figures, namely the use of different colors in the column graphs for the different testosterone concentrations tested.Resp: the authors agreed and made the change of the colors in the grapphics Response Reviewer 2: The authors agree and a small modification of the different figures, namely the use of different colors in the column graphs for the different testosterone concentrations tested was done. |
|
|
Reviewer 2 Report
Comments and Suggestions for Authors
The manuscript entitled "Testosterone Nanoemulsion prevents Prostate Cancer: PC-3 and LNCaP Cell Viability in Vitro" is an interesting research work that paves the way for new possible therapeutic strategies in the treatment of prostate cancer. With this study the authors demonstrate that, contrary to what has been hypothesized so far, the supplementation of nanoemulsified testosterone leads to a reduction in the metabolic activity and viability of prostate cancer cells.
However, I would suggest to the authors a small modification of the different figures, namely the use of different colors in the column graphs for the different testosterone concentrations tested.
Author Response

(The authors gave the same response as above.)

Round 2
Reviewer 1 Report
Comments and Suggestions for Authors
I think the authors revised the manuscript reasonably.
Author Response
For research article
Testosterone Nanoemulsion prevents Prostate Cancer: PC-3 and LNCaP Cell Viability in Vitro
|
Response to Reviewer X Comments
|
||
|
1. Summary |
|
|
|
Thank you very much for taking the time to review this manuscript. Please find the detailed responses below and the corresponding revisions/corrections highlighted/in track changes in the re-submitted files. |
||
|
|
|
|
|
|
|
|
|
|
|
|
|
|
|
|
|
|
|
|
|
|
|
|
|
|
|
|
|
2. Point-by-point response to Comments and Suggestions for Authors |
||
|
Comments 1: - The testoterone concentrations used are extremely high, far above the |
||
|
Response 1: Thank you for pointing this out. I agree with this comment. Therefore, i have to explain that we have proceed the same amounts of testosterone concentrations performed as other authors have tested. In our study testosterone was tested in different range concentrations from 62.5 - 250 µM as used by different authors evaluating testosterone activity in this model, this is a regular and normal concentrations tested by other published papers around the world, thus we think that The testoterone concentrations used in this paper are normal and this is not high, far above the biologically levels. The concentration is suitable to test biolologically as recommended as regular concentrations used in many articles testing testosterone as you can see below including the tests performed by ECHA (European Chemicals Agency) to test the drugs commercialized in the actual Market.
Published Articles using the same concentrations
Article 1 Drug Metab Dispos. 2017 Dec; 45(12): 1266–1275. Digging Deeper into CYP3A Testosterone Metabolism: Kinetic, Regioselectivity, and Stereoselectivity Differences between CYP3A4/5 and CYP3A7 Published online 2017 Dec. doi: 10.1124/dmd.117.078055 PMCID: PMC5697443 PMID: 28986474
The incubation reactions contained various concentrations of testosterone (2.5–500 µM) that were dissolved in methanol (1% v/v);
Article 2 Effects of Dietary Components on Testosterone Metabolism via UDP-Glucuronosyltransferase Front Endocrinol (Lausanne). 2013; 4: 80. Published online 2013 Jul 8. Prepublished online 2013 May 19. doi: 10.3389/fendo.2013.00080 PMCID: PMC3703584 PMID: 23847592
testosterone glucuronidation by 72, 22, and 9% respectively, with concentrations of phenolic: testosterone of 100 μM: 250 μM.
Article 3
Modulating testosterone pathway: a new strategy to tackle male skin aging? Philippe Bernard,1 Thomas Scior,2 and Quoc Tuan Do1 Clin Interv Aging. 2012; 7: 351–361. Published online 2012 Sep 13. doi: 10.2147/CIA.S34034 PMCID: PMC3459575 PMID: 23049247
Test compounds were dissolved in dimethyl sulfoxide (DMSO) and serially diluted in Dulbecco’s modified Eagle’s medium to reach the three final test concentrations of 100 μM.
European Chemicals Agency - ECHA Administrative dataEndpoint: additional toxicological information Type of information: experimental study Adequacy of study: key study Study period:October-January 2018 Reliability: 1 (reliable without restriction) Rationale for reliability incl. deficiencies: https://echa.europa.eu/registration-dossier/-/registered-dossier/13353/7/13 guideline study Statistical analysis showed that 1,2,4-Triazole significantly (p =0.05) decreased testosterone release in H295R cells at a concentration of 0.1 µM and significantly increased testosterone release at a concentration of 100 µM.
Comments 2: A CETSA analysis would be essential to identify which proteins are actually bound in presence of these extremely high hormone concentrations.
Response 2: Thank you for pointing this out. We agree with this comment. Therefore, we have to explain that Although CETSA analysis enables target engagement studies without requiring modifications to the proteins or compounds of interest. We believe that choosing the flow citometry analisys and the alamar blue tests, they both provides reliable, strong and significant information for the puspose of this study.
Comments 3: In any case, the biologically much more relevant androgen form is DHT, which activates the AR at subnanomolar concentrations. Response 3: Thank you for pointing this out. We agree with this comment. Therefore, we have to explain that we used testosterone since DHT is not used as a compound for prostate cancer in Brazil, but testosterone in cream is largely used.
Comments 4: Comparative data would be needed. How is this explained?
Comments 5: - It is unclear and not really discussed why nanoemulsions were used for
Response 5: Thank you for pointing this out. We agree with this comment. Therefore, we have to explain that we tested the nanoemulsion because in Brazil they are used in gel form and in Emulsion, thus this article would be very important to be published since the gel form has been used for many yeas with poor results since its is derived from carbopol a water derived with very small amount of transcutaneous permeability.
|
||
|
|
||
|
|
||
|
|
||
|
|
||
|
|
||
|
5. Additional clarifications |
||
|
We have been using testosterone emulsion in Brasil for many years and the results are published elsewhere with impactant results for breas cancer prevention in neopausal women in longitudinal studies such as listed below:
Botelho MA, Queiroz DB, Barros G. Nanostructured transdermal hormone replacement therapy for relieving menopausal symptoms: a confocal Raman spectroscopy study. Clinics (Sao Paulo). 2014 Feb;69(2):75-82. doi: 10.6061/clinics/2014(02)01. PMID: 24519196; PMCID: PMC3912337.
The effect of bioidentical nanostructured progesterone in the in vitro culture of preantral follicles and oocyte maturation. Neto CC, Soares KL, Padilha RT, Botelho MA, Queiroz DB, Figueiredo JR, de Melo Magalhães-Padilha D.Cell Tissue Res. 2020 Dec;382(3):657-664. doi: 10.1007/s00441-020-03233-6. Epub 2020 Jul 21.PMID: 32696218
Effects of a transdermal testosterone metered-dose nanoemulsion in peri- and postmenopausal women: a novel protocol for treating low libido MedicalExpress (São Paulo, online) 2 (5) • Oct 2015 • https://doi.org/10.5935/MedicalExpress.2015.05.03 https://www.scielo.br/j/medical/a/6fw6YGYnbwXZv5b4Z8x6pJk/abstract/?lang=en
Botelho MA, Queiroz DB, Freitas A, et al. Effects of a new testosterone transdermal delivery system, Biolipid B2-testosterone in healthy middle aged men: a Confocal Raman Spectroscopy Study. J Pharm Sci Innov. 2013;2(2):1–7. 10.7897/2277-4572.02204
Gonzaga LW, Botelho MA, Queiroz DB. Nanotechnology in Hormone Replacement Therapy: Safe and Efficacy of Transdermal Estriol and Estradiol Nanoparticles after 5 Years Follow-Up Study. Lat Am J Pharm. 2012; 31: 442-450.
|
||
